# Model-Based Observer Design Considering Unequal Measurement Delays

Yousef Alipouri and Lexuan Zhong *

Department of Mechanical Engineering, University of Alberta, Edmonton, AB T6G 2V4, Canada;
alipouri@ualberta.ca
* Correspondence: lexuan.zhong@ualberta.ca

**Abstract:** State observers are essential components of a modern control system. It is often designed based on a mathematical model of the process, thus requiring detailed process knowledge. However, in the existing state estimation methods, equal delays are commonly assumed for all communication lines, which is unrealistic and poses problems such as instability and a degraded performance of observers when unequal time delays exist. In this paper, a design of observers considering the measurement delays is presented. To deal with this problem, a chain-based observer has been proposed in which each chain deals with one output delay, performs prediction for the unavailable output value, and passes it to the next chain. Convergence of each chain observer as well as overall state estimation were proven. To illustrate the performance of the proposed scheme, simulation studies were performed on a benchmark continuous stirred tank heater (CSTH) process.

**Keywords:** model based observer; chain observer; measurement delays; continuous stirred tank heater; state estimation



## 1. Introduction

State observers play a key role in process monitoring, control, and automation by providing estimates of internal state variables using hardware output measurements. Given a process model, observers can be designed by a large number of available methods such as high gain [1], Kalman filter [2], sliding mode [3], and so on. Various types of observers have also been used in the feedback control design in the presence of delays [4,5]. One of the profound challenges in state estimation is that sensor measurements are often time delayed from the actual process outputs. The control problem of dynamic systems with delays has received growing attention in recent years due to novel application areas such as control over networks [6]. This might also occur due to the slow dynamics of physical sensors as well as communication delays from measurements transmitted between different processing units. If these delays are not compensated, they can lead to the performance degradation of the observers and can have a strong impact on their stability and robustness properties [7,8]. When designing an observer or controller for a system affected by such delays, it is often expedient to lump the measurement delay together with any system and actuation delay, as they are indistinguishable in terms of the ideal closed-loop dynamics of the system [9]. However, in some cases, it can be advantageous to model the measurement delays separately from the other delays.

For linear systems, when the system model is available, there exists literature to deal with equal output delays in state estimation [10–12]. Assuming equal delays for all output channels is unrealistic due to different communication line lengths or different sensors and technology types. Hence, the involvement of the delay could be in the communication line, be multiple, and different for each channel, which, however, has rarely been studied in the literature. Supposing equal delays will simplify the observer designing step, but can cause similar problems to ignoring them [13]. Others have assumed the delays on outputs [9,14] or states [15], which again, is simpler than considering them in the communication line (out

of the process model). Furthermore, an extension of the available model-based methods to data-driven methods is not straightforward (if even it is possible). This paper intends to extend the Luenberger observer design proposed in [16] to account for equal and unequal measurement delays. Then, the subspace-based approach [17] can similarly be extended to design a data-driven observer in the presence of equal and unequal communication delays.

In this paper, we assumed that the output variables were measured but with delays. Furthermore, the delays might be different among the different output variables. Thus, the receiver node needs to be reconstructed to be compatible with the available data. To deal with this problem, a chain observer has been proposed in which each chain collects data, performs prediction, and passes it to the next chain. The chain observer is a common method to deal with delays in nonlinear systems. The idea of achieving the convergence of the state estimate by using a cascade of two observers (elementary chain observer) for a nonlinear system was first proposed in [18], while the idea of using more than two observers in the chain to deal with large measurement delays was proposed in [19]. In [18], it was shown that a chain of two observers is sufficient for asymptotic state reconstruction as long as the measurement delay is below a given threshold, which depends on the Lipschitz constants of the system. When the time delay exceeds such a threshold, more links must be added to the chain [19]. Thus, each observer in the chain is in charge of predicting the system state for a suitable fraction of the total delay. The chain observer method has also been used for dealing with large measurement delays [7].

In the proposed method, the chain observer was used to deal with the multiple unequal measurement delays. It was proven that the proposed chain observer had the desired stability properties with delay-free measurements, and the design followed a two-step strategy. First, it obtained the delayed state trajectories using delayed measurements with the delay-free observer. These delayed state estimates were then used as predictors that can compensate for the presence of delay. This strategy was shown to converge to true states. In this paper, the convergence of each chain as well as the overall observer was proven. This paper shows the development of the chain observer, which holds stability as well as convergence, along with a new prediction method that is used in each chain.

The main contributions of this paper are as follows:

1. Designing an observer by considering equal measurement delays.
2. Designing a chain observer to deal with unequal measurement delays.
3. Proving the convergence of each chain as well as the overall observer.

The rest of this paper is organized as follows. In Section 2, the preliminaries are presented. The problem formulation is discussed in Section 3. Section 4 presents the theory of the proposed Luenberger-based observer for the case of equal and unequal measurement delays. In Section 5, a simulation through the continuous stirred tank heater (CSTH) process is used to illustrate the performance of the proposed scheme. The paper ends with concluding remarks in the final section.

## 2. Preliminaries of Model-Based (Luenberger) Observer Design

In this section, we present the preliminaries of designing a model-based observer approach following the work of [17]. Consider a discrete-time LTI (linear time invariant) system, which is described by

$$\begin{aligned} x(k+1) &= Ax(k) + Bu(k) \\ y(k) &= Cx(k) + Du(k) \end{aligned} \tag{1}$$

where $u(k) \in \mathrm{R}^l$, $y(k) \in \mathrm{R}^m$, and $x(k) \in \mathrm{R}^n$ represent process input, output, and state variable vectors, respectively. Suppose that the pair $(C, A)$ with $A \in \mathrm{R}^{n \times n}$, $C \in \mathrm{R}^{m \times n}$ is observable. The s $(= n)$-order observer with the following state and output equations

$$z(k+1) = Gz(k) + Hu(k) + Ly(k) \in \mathrm{R}^s \tag{2}$$

$$\hat{y}(k) = Wz(k) + V\,y(k) + Qu(k) \in \mathrm{R}^m \tag{3}$$

is asymptotically stable, provided that $G$ is stable (all of its eigenvalues lie within the circle one) and the observer parameters $H, L, Q, W, V$ satisfy the following Luenberger equations:

$$TA - GT = LC, H = TB - LD$$
$$C = WT + VC, Q = D - VD \tag{4}$$
$$G \in \mathrm{R}s \times s, T \in \mathrm{R}s \times n, W \in \mathrm{R}m \times s$$

where $z(k)$ is the observer states related to the process states through the transformation matrix $T \in \mathrm{R}^{s \times n}$ so that $z(k) = T\hat{x}(k)$, where $\hat{x}(k)$ is an estimation of true state $x(k)$.

### 3. Problem Formulation

Sensors and communication networks are an integral part of a data transmission system. One of the main issues regarding measurements and network transmission is the presence of delays in each output measurement (say delay $\tau_i$ for output $y_i$). Figure 1 illustrates the communication network and corresponding delays. Due to these delays, output signals $y(k)$ required for state estimation (2) are not available instantaneously. Therefore, the objective of this work was to design an observer that utilizes delayed measurements (i.e., $y(k - \tau) = [y_1(k - \tau_1) \ldots y_m(k - \tau_m)]$) to estimate $z(k)$. As a special case, these delays may be equal in all outputs (i.e., $\tau_i = d$, for $i = 1, \ldots, m$.) First, we developed an efficient observer scheme in the presence of *constant equal* measurement delays. The proposed observer, which deals with equal measurement delays, will have the form of:

$$z(k) = Gz(k - d) + \sum_{i=0}^{d-1} H_i u(k - 1 - i) + Ly(k - d) \tag{5}$$
$$T\hat{x}(k) = z(k)$$

where it utilizes available data until the $(k - d)^{th}$ instant to estimate $x(k)$. The next section will also propose a novel method to deal with unequal measurement delays using a chain–observer structure.

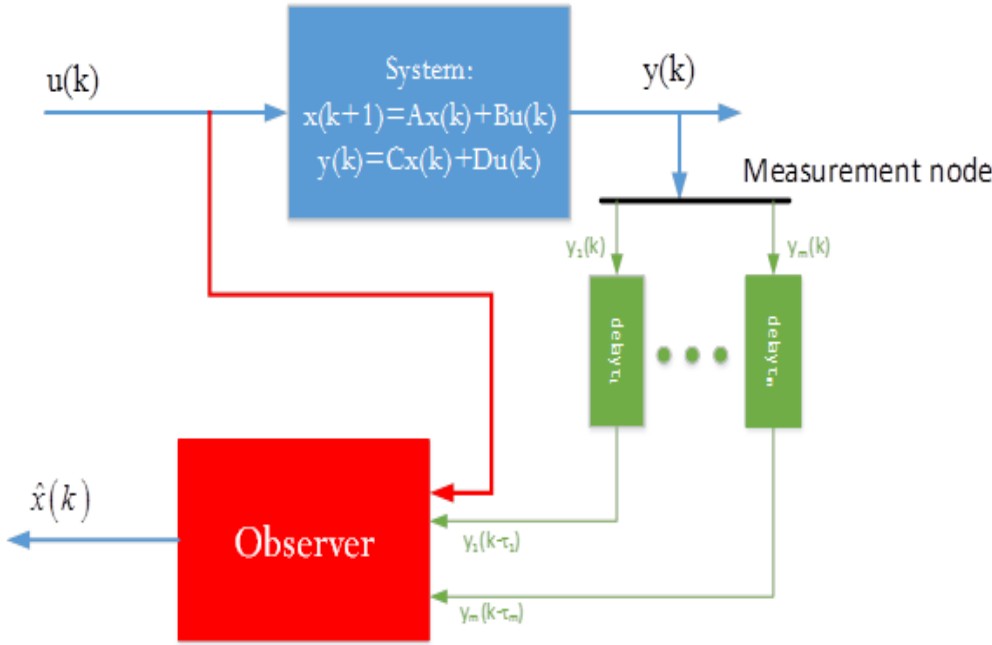

**Figure 1.** Schematic showing the transmission lines with measurement delays.

## 4. The Proposed Observer Design

*4.1. Observer Design in the Case of Equal Measurement Delays*

In this section, it was assumed that the delays were equal in all outputs (i.e., $\tau_i = d$, for $i = 1, \ldots, m$). The following theorem designs an asymptotically stable observer (5) with the consideration of equal measurement delays when the system model is available. It is an extension of the Luenberger condition (4) for the case of equal measurement delays.

**Theorem 1.** *Suppose system model (1) is known with equal measurement delay d in all measured outputs. Furthermore, it is assumed that the pair (C,A) is observable. The observer with the following state and output equations*

$$z(k) = Gz(k-d) + \sum_{i=0}^{d-1} H_i u(k-1-i) + Ly(k-d) \in R^s$$
$$T\hat{x}(k) = z(k) \tag{6}$$

$$\hat{y}(k) = Wz(k-d) + Vy(k-d) + \sum_{i=0}^{d} Q_i u(k-i) \tag{7}$$

*is asymptotically stable, namely $\lim_{k\to\infty} (Tx(k) - z(k)) \to 0$, and has unbiased estimation for $y(k)$, that is $\lim_{k\to\infty} (\hat{y}(k) - y(k)) = 0$, provided the observer parameters defined in Equations (6) and (7) satisfy the following delayed-Luenberger equations:*
*State convergence condition:*

$$TA^d - GT = LC, \; H_{d-1} = TA^{d-1}B - LD,$$
$$H_i = TA^i B, \; i = 0, \ldots, d-2 \tag{8}$$

*Output convergence condition:*

$$CA^d = WT + VC, \; Q_0 = D - VD,$$
$$Q_j = CA^{j-1}B, \; j = 1,\ldots, d \tag{9}$$
$$G \in Rs \times s, \; T \in Rs \times n$$

*where G should be chosen so that it is stable (i.e., every eigenvalue of G lies within the circle one). In addition, the closed loop dynamics of estimation error for state and output is reduced to*

$$e(k) = Tx(k) - z(k)$$
$$e(k) = Ge(k-d) \tag{10}$$
$$y(k) - \hat{y}(k) = We(k-d).$$

**Proof.** The state equation of Equation (1) can be rewritten as:

$$x(k) = Ax(k-1) + Bu(k-1) \tag{11}$$

Substituting $x(k)$ from (11) in (1) yields

$$(k+1) = A^2 x(k-1) + Bu(k) + ABu(k-1) \tag{12}$$

Similarly, $x(k-1)$ can be replaced by $Ax(k-2) + Bu(k-2)$, which yields

$$x(k+1) = A^3 x(k-2) + Bu(k) + ABu(k-1) + A^2 Bu(k-2) \tag{13}$$

Continuing the above procedure, after d step reaches

$$(k+1) = A^{d+1} x(k-d) + \sum_{i=0}^{d} A^i Bu(k-1-i) \tag{14}$$

or

$$x(k) = A^d x(k-d) + \sum_{i=0}^{d-1} A^i Bu(k-1-i) \tag{15}$$

The state estimation error is defined as

$$(k) = Tx(k) - z(k) \tag{16}$$

Substituting Equations (15) and (6) in Equation (16) gives

$$\begin{aligned} e(k) = &\; TA^d x(k-d) + \sum_{i=0}^{d-1} TA^i Bu(k-1-i) \\ &- Gz(k-d) - \sum_{i=0}^{d-1} H_i u(k-1-i) - Ly(k-d) \end{aligned} \tag{17}$$

Now by substitution of $TA^d - GT = LC$, $H_{d-1} = TA^{d-1}B - LD$, $H_i = TA^i B$, and $y(k-d) = Cx(k-d) + Du(k-d)$, the above equation is reduced to

$$(k) = Ge(k-d) \tag{18}$$

The estimation error (18) will converge to zero, provided $G$ is a stable matrix. Performing similar steps for $\hat{y}(k) - y(k)$ gives

$$y(k) - \hat{y}(k) = We(k-d) \tag{19}$$

Converging $e(k)$ to zero proves relation (10). Thus, the proof of Theorem 1 is completed. $\square$

**Remark 1.** *The matrix G can be defined as follows:*

$$G = \begin{bmatrix} G_0 & g \end{bmatrix}, \quad g = \begin{bmatrix} g_1 \\ \vdots \\ g_{s-1} \\ g_s \end{bmatrix}, \quad G_0 = \begin{bmatrix} 0 & 0 & \cdots & 0 \\ 1 & 0 & \cdots & 0 \\ \vdots & \ddots & \ddots & \ddots \\ 0 & \cdots & 1 & 0 \\ 0 & \cdots & 0 & 1 \end{bmatrix} \tag{20}$$

*then, the designed observer poles are roots of polynomial function $x_s - g_s x_{s-1} - \cdots - g_1 = 0$. Therefore, to have the desired performance, one can select proper poles and determine the corresponding polynomial function coefficients ($g_i$ for $i = 1,\ldots, s$).*

### 4.2. Observer Design in the Case of Unequal Delays

This section considers the extension of the proposed observers (6) and (7) to deal with unequal measurement delay. The proposed observer consists of $m$ chain–observers, each one being in the form of (6) and (7). Without loss of generality, let $\tau_1 \leq \tau_2 \leq \cdots \leq \tau_m$ and define $d_1 = \tau_1$, $d_i = \tau_i - \tau_{i-1}$, $i = 2,\ldots,m$. Before explaining the proposed approach of designing the observer considering unequal delay, a theorem that provides stability conditions for a system with multiple delays is presented. This theorem is an extension to the single delay case, which was presented in [20]. The result of this theorem enables us to design an observer with the convergence property.

**Theorem 2.** *Suppose the system model with the assumption $\|A_0\|_2 \neq 0$, is given by*

$$x(k+1) = A_0 x(k) + \sum_{j=1}^{N} A_j x(k-d_j) \tag{21}$$

*where N is the value of time delays. Then, it is asymptotically stables if there exists a real symmetric matrix $P > 0$ so that:*

$$(1 + \varepsilon_m) A_0^T P A_0 + N\left(\frac{1+\varepsilon_m}{\varepsilon_m}\right) \sum_{j=1}^{N} A_j{}^T P A_j - P < 0$$

$$\varepsilon_m = \left(N \sum_{j=1}^{N} \|A_j\|_2^2\right)^{\frac{1}{2}} \|A_0\|_2^{-1} \tag{22}$$

*where* $\|\Psi_2\| = \sqrt{\lambda_{\max}(\Psi^T \Psi)}$

**Proof.** See Appendix A. □

**Remark 2.** *When $\|A_0\|_2 = 0$, system model (21) can be rewritten as*

$$x(k+1) = \sum_{j=1}^{N} A_j x(k - d_j)$$

$$= A_1 x(k - d_1) + \sum_{j=2}^{N} A_j x(k - d_j) \tag{23}$$

*Now, define a new variable as $x'(k) = x(k - d_1)$. Then, the augmented model becomes*

$$\begin{bmatrix} x(k+1) \\ \acute{x}(k+1) \end{bmatrix} = \begin{bmatrix} 0 & 0 \\ 0 & A_1 \end{bmatrix} \begin{bmatrix} x(k) \\ \acute{x}(k) \end{bmatrix} + \sum_{j=2}^{N} \begin{bmatrix} A_j & 0 \\ 0 & 0 \end{bmatrix} \begin{bmatrix} x(k - d_j) \\ \acute{x}(k - d_j) \end{bmatrix} \tag{24}$$

*or*

$$\overline{x}(k+1) = \overline{A}_0 \overline{x}(k) + \sum_{j=2}^{N} \overline{A}_j \overline{x}(k - d_j) \tag{25}$$

*where* $\overline{x}(k) = \begin{bmatrix} x(k) \\ \acute{x}(k) \end{bmatrix}$, $\overline{A}_0 = \begin{bmatrix} 0 & 0 \\ 0 & A_1 \end{bmatrix}$, *and* $\overline{A}_j = \begin{bmatrix} A_j & 0 \\ 0 & 0 \end{bmatrix}$. *Then, Theorem 2 can be applied for stability analysis of (25).*

Suppose that due to different sensors or measurement techniques or communication topologies, each output measurement is received with a delay that may be different from delays in other output channels. Therefore, in order to estimate the state $x(k)$ at sample time $k$, the output $y_i$ is only available up to $k - \tau_i$ for $i = 1, \ldots, m$. The observer gains can be parameterized by the set of the available measurements. In order to handle the unequal measurement delays, we propose constructing a new observer scheme. The proposed observer scheme is shown in Figure 2, in which $y_i(1 : s)$ is the output $i$ from sample 1 to $s$. The proposed observer gains can be calculated in a composition of $m$ chain–observers, each of which predicts the unavailable output due to measurement delay as well as updates the values of the estimated states and then passes them to the next chain.

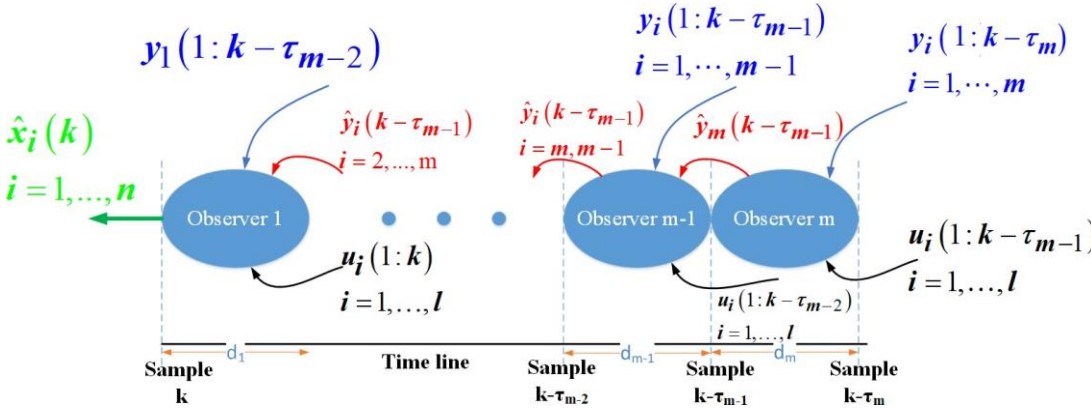

**Figure 2.** The proposed *m* chain–observer ($y_i$ (1: *s*) is the output *i* from sample 1 to sample *s*).

**Theorem 3.** *Suppose that A, B, C, D, in Equation (1) are known, and $\tau = [\ \tau_1 \ \cdots \ \tau_m]$ represents unequal measurement delays, and define $d_1 = \tau_1$, $d_i = \tau_i - \tau_{i-1}$, $i = 2, \ldots, m$. The design of m-chain–observer is achieved by solving the chain-Luenberger equations for $i = 1, \ldots, m$ chain–observers. The observer i is designed using the estimated output $\hat{y}_{j+1}(k - \tau_i)$ for $j = i, \ldots, m - 1$ provided by the previous chain–observers from 1 to $i - 1$ as*

$$
\begin{aligned}
z(k - \tau_{i-1}) = G_i z(k - \tau_i) + L_i \bar{y}(k - \tau_i) \\
+ \sum_{i=0}^{d_i - 1} H_i u(k - 1 - i - \tau_{i-1})
\end{aligned}
\tag{26}
$$

*where*

$$
\hat{y}(k - \tau_{i-1}) = [y_1(k - \tau_i) \ \cdots \ y_i(k - \tau_i) \ \hat{y}_{i+1}(k - \tau_i) \ \hat{y}_m(k - \tau_i) \ \hat{y}_m(k - \tau_i)]
$$
$$
T_i \hat{x}(k - \tau_{i-1}) = z(k - \tau_{i-1})
$$

*and*

$$
\begin{aligned}
\hat{y}(k - \tau_{i-1}) = W_i z(k - \tau_i) + V_i \bar{y}(k - \tau_i) \\
+ \sum_{i=0}^{d_i} Q_i u(k - i - \tau_{i-1})
\end{aligned}
\tag{27}
$$

*Each chain–observer satisfies the following convergence conditions:*

$$
T_i A^{d_i} - G_j T_i = L_i C, \quad H_{d_i - 1} = T_i A^{d_i - 1} B - L_i D
\tag{28}
$$

*where $G_i$ for $i = 1, \ldots, m$ should be chosen so that it is stable and should satisfy the stability condition Theorem 2 as*

$$
(1 + \varepsilon_i) G_i^T P G_i + (m - i) \left( \frac{1 + \varepsilon_i}{\varepsilon_i} \right) \sum_{j=i+1}^{m} \Phi_j^T P \Phi_j - P < 0
$$
$$
\varepsilon_i = \left( (m - i) \sum_{j=i+1}^{m} \|\Phi_j\|_2^2 \right)^{\frac{1}{2}} \|G_i\|_2^{-1}
\tag{29}
$$

*where $P > 0$, $\Phi_j = -L_i \prod_{p=i+1}^{m} V_p W_j I_j$. It follows then, that the constructed observer is asymptotically convergent, and the observer output equation delivers unbiased estimation for $y(k)$, that is,*

$$
\lim_{k \to \infty} (\hat{y}(k) - y(k)) \to 0
\tag{30}
$$

**Proof.** See Appendix B. □

To design an observer with unequal output delay, Theorem 3 can be utilized to construct each chain observer provided that $g_i$ (i.e., $G_i$) in each chain must be selected in such a way that the stability condition (29) is satisfied. Therefore, the chain observer can be designed using the following algorithm (Algorithm 1):

---

**Algorithm 1.** Chain observer design algorithm

---

**Step 1:** set $i = m$
**Step 2:** Select a proper $G_i$ that satisfies stability conditions (29).
**Step 3:** Design the chain-observer $i$ using Theorem 1 with $d = d_i$.
**Step 4:** Produce the complementary output $\overline{y}(k - \tau_{i-1})$ using available output and input, then estimate output data by Equation (27).
**Step 5:** Go to **step 2** and repeat till $i = 1$.

---

## 5. Simulation

In this section, the effectiveness of the proposed approach was demonstrated on a simulation setup of a continuous stirred tank heater (CSTH). The system is written as a linear state-space model to generate input–output data with the following parameters [16]:

$$x = \begin{bmatrix} V_T \\ H_T \\ T_{hj} \end{bmatrix}, \quad y = \begin{bmatrix} h_T \\ T_T \\ T_{hj} \end{bmatrix} \tag{31}$$

$$A = \begin{bmatrix} 0 & 0 & 0 \\ -626.4371 & -5.9406 \times 10^{-3} & 36.55 \\ 0 & 0 & -1.2019 \times 10^{-3} \end{bmatrix}$$

$$B = \begin{bmatrix} 1 & 0 \\ 0 & 0 \\ 0 & 3041.5241 \end{bmatrix} \tag{32}$$

$$C = \begin{bmatrix} 31.831 & 0 & 0 \\ 0 & 3.8578 \times 10^{-5} & 0 \\ 0 & 0 & 1 \end{bmatrix}$$

where $\dot{V}_{in} - \dot{V}_{out} \equiv u_1$, $P_h \equiv u_2$ are two input variables, and the water level $h_T$, the temperature of the water in the tank $T_T$ as well as in the heating jacket $T_{hj}$ are output variables. The physical meanings of the process variables and parameters used above are listed in Table 1.

**Table 1.** The process variables and parameters.

| Symbol | Description | Unit |
|---|---|---|
| $V_T$ | Water volume in the tank | L |
| $H_T$ | Enthalpy in the tank | J |
| $T_{hj}$ | Temperature in the heating jacket | °C |
| $\dot{V}_{in}, \dot{V}_{out}$ | Water flows in and out of the tank | 1/s |
| $P_h$ | Electrical heater power | W |
| $h_T$ | Water level in the tank | m |
| $T_T$ | Water temperature in the tank | °C |

The output variables had the delay vector of $\tau = [2\ 5\ 7]$. The order of the observer was $s = 3$. Set $g_1 = [10^{-2}, -3 \times 10^{-8}, 3 \times 10^{-4}]$ so that all three poles were located in $10^{-4}$ at each chain. Then, we designed a chain–observer using Algorithm 1 when $m = 3$.

The following is the design procedure:
**Chain 1-**
Step 1.1: Set $d = 7 - 5 = 2$ and $s = 3$;

Step 1.2: Set

$$g = \begin{bmatrix} 10^{-12} \\ -3 \times 10^{-8} \\ 3 \times 10^{-4} \end{bmatrix} \tag{33}$$

$$G = \begin{bmatrix} 0 & 0 & 10^{-12} \\ 1 & 0 & -3 \times 10^{-8} \\ 0 & 1 & 3 \times 10^{-4} \end{bmatrix}$$

which results in three poles at $10^{-4}$.

Step 1.3: Calculate $L$, $T$, $H_i$ as

$$L = \begin{bmatrix} -3 \times 10^{-7} & -2.3 \times 10^{-6} & 8.3 \times 10^{-12} \\ -1.9 \times 10^{-4} & 6.7 \times 10^{-2} & -5.7 \times 10^{-6} \\ 0.02 & 0.0172 & -1.2 \times 10^{-6} \end{bmatrix} \tag{34}$$

$$H_1 = \begin{bmatrix} 1.6 \times 10^{-3} & -2.1 \times 10^{-4} \\ 4.1 \times 10^{-4} & -4.5 \times 10^{-5} \\ -1.3 \times 10^{-4} & 2.4 \times 10^{-5} \end{bmatrix}$$

$$T = \begin{bmatrix} 6.2 \times 10^{-3} & -2.6 \times 10^{-6} & 5.7 \times 10^{-6} \\ -0.652 & -6.6 \times 10^{-7} & -1.2 \times 10^{-6} \\ -0.732 & 2.1 \times 10^{-7} & -6.7 \times 10^{-7} \end{bmatrix}$$

$$H_0 = \begin{bmatrix} 6.21.6 \times 10^{-3} & 0.175 \\ -0.652 & 0.038 \\ -0.7323 & -0.02 \end{bmatrix}$$

Step 1.4: Calculate $V$, $W$, $Q_i$ as

$$W = \begin{bmatrix} -0.32 & 1 & -0.89 \\ 162.21 & -507.3 & 452.98 \\ 246.22 & -769.4 & 687.04 \end{bmatrix} \tag{35}$$

$$V = \begin{bmatrix} 2 \times 10^{-12} & -1.1 \times 10^{-7} & 2.3 \times 10^{-15} \\ 4.5 \times 10^{-6} & 3.5 \times 10^{-5} & 0 \\ -1.4 \times 10^{-9} & 8.2 \times 10^{-5} & 1.4 \times 10^{-6} \end{bmatrix}$$

$$V = \begin{bmatrix} 2 \times 10^{-12} & -1.1 \times 10^{-7} & 2.3 \times 10^{-15} \\ 4.5 \times 10^{-6} & 3.5 \times 10^{-5} & 0 \\ -1.4 \times 10^{-9} & 8.2 \times 10^{-5} & 1.4 \times 10^{-6} \end{bmatrix}$$

$$Q_0 = \begin{bmatrix} 0 & 0 \\ 0 & 0 \\ 0 & 0 \end{bmatrix}$$

$$Q_1 = \begin{bmatrix} 31.83 & 0 \\ 0 & 0 \\ 0 & 30411.5 \end{bmatrix}$$

$$Q_2 = \begin{bmatrix} 0 & 0 \\ -0.0241 & 0 \\ 0 & -36.55 \end{bmatrix}$$

**Chain 2-**

Step 2.1: Set $d = 5 - 2 = 3$ and $s = 3$;

Step 2.2: Set

$$g = \begin{bmatrix} 10^{-12} \\ -3 \times 10^{-8} \\ 3 \times 10^{-4} \end{bmatrix} \tag{36}$$

$$G = \begin{bmatrix} 0 & 0 & 10^{-12} \\ 1 & 0 & -3 \times 10^{-8} \\ 0 & 1 & 3 \times 10^{-4} \end{bmatrix}$$

which results in three poles at $10^{-4}$.

Step 2.3: Calculate $L$, $T$, $H_i$

$$L = \begin{bmatrix} -2.5 \times 10^{-8} & 2 \times 10^{-4} & -1.4 \times 10^{-11} \\ -9.2 \times 10^{-4} & 0.998 & 8.27 \times 10^{-11} \\ -1.6 \times 10^{-4} & 0.0236 & -1 \times 9 \end{bmatrix} \tag{37}$$

$$H_2 = \begin{bmatrix} -1.4 \times 10^{-4} & -3.1 \times 10^{-7} \\ -3 \times 10^{-6} & -4 \times 10^{-7} \\ 6.4 \times 10^{-7} & -3.3 \times 10^{-7} \end{bmatrix}$$

$$T = \begin{bmatrix} 2.9 \times 10^{-2} & -3.8 \times 10^{-5} & -8.2 \times 10^{-11} \\ 5.3 \times 10^{-3} & -9.1 \times 10^{-7} & 1 \times 10^{-9} \\ -4.5 \times 10^{-5} & 8.7 \times 10^{-8} & 1.2 \times 10^{-9} \end{bmatrix}$$

$$H_1 = \begin{bmatrix} 0.0241 & -3.2 \times 10^{-8} \\ 5.7 \times 10^{-4} & -3.8 \times 10^{-7} \\ -5.4 \times 10^{-5} & -4.6 \times 10^{-7} \end{bmatrix}$$

$$H_0 = \begin{bmatrix} 0.0295 & -2.4 \times 10^{-6} \\ 5.3 \times 10^{-3} & 3 \times 10^{-5} \\ -4.6 \times 10^{-5} & 3.6 \times 10^{-5} \end{bmatrix}$$

Step 2.4: Calculate $V$, $W$, $Q_i$ as

$$W = \begin{bmatrix} 0.1 & -0.53 & 6.35 \\ 0.007 & -0.036 & 0.428 \\ -1.48 & 7.51 & -87.86 \end{bmatrix} \tag{38}$$

$$V = \begin{bmatrix} -1 \times 10^{-11} & -3 \times 10^{-6} & 2.3 \times 10^{-15} \\ -2.6 \times 10^{-8} & -2.1 \times 10^{-7} & 5 \times 10^{-16} \\ 1.5 \times 10^{-10} & 4.3 \times 10^{-5} & -1.7 \times 10^{-9} \end{bmatrix}$$

$$Q_0 = \begin{bmatrix} 0 & 0 \\ 0 & 0 \\ 0 & 0 \end{bmatrix}$$

$$Q_1 = \begin{bmatrix} 31.83 & 0 \\ 0 & 0 \\ 0 & 30411.5 \end{bmatrix}$$

$$Q_2 = \begin{bmatrix} 0 & 0 \\ -0.0241 & 0 \\ 0 & -36.55 \end{bmatrix}$$

$$Q_3 = \begin{bmatrix} 0 & 0 \\ 1.4 \times 10^{-4} & 0 \\ 0 & 0.044 \end{bmatrix}$$

**Chain 3**-Same as **Chain 1**.

The designed chain–observer (Method 1) was compared with the classic observer (Method 3), which ignores the measurement delays, and the observer (Method 2) [21], which supposes (one) equal measurement delay (delay = 2 in this test). The method proposed in [21] deals with both state and output delays; in this test, state delay is supposed to be equal to zero. Figures 3–5 show the estimated values for the system states $x_1$, $x_2$, and

$x_3$ by the above-mentioned observers, respectively. These figures show that the proposed observer estimation error converges to zero in 10 s. Observer #2 (considered equal delays) had a bias in estimation and was slow in convergence, and observer #3 (ignored delays) diverged. The input signal to the system is depicted in Figure 6.

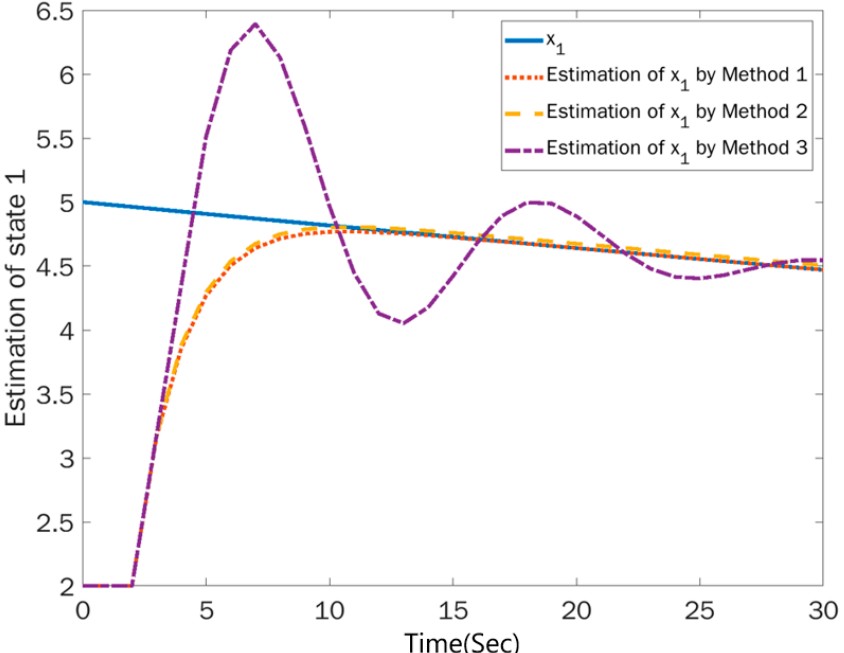

**Figure 3.** The state 1 estimation results of the three introduced observers.

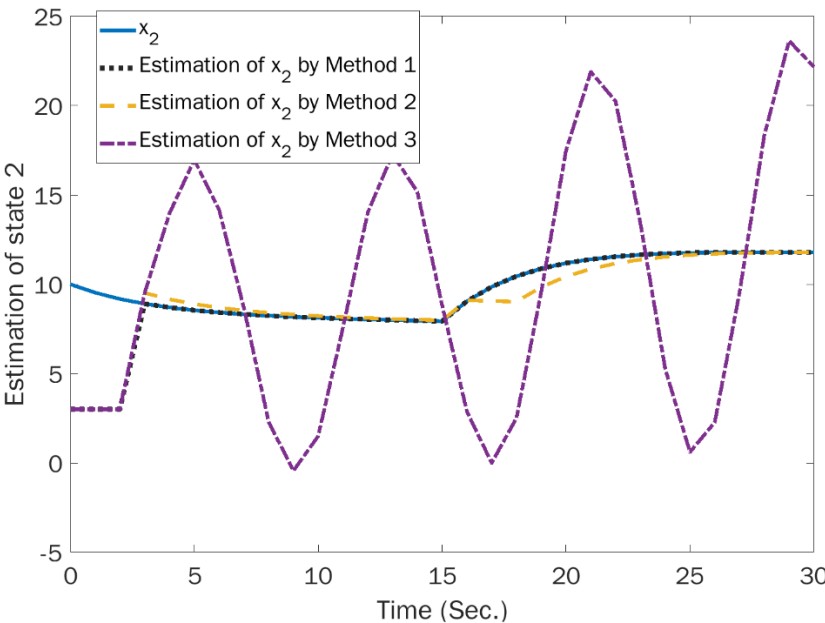

**Figure 4.** The state 2 estimation results of the three introduced observers.

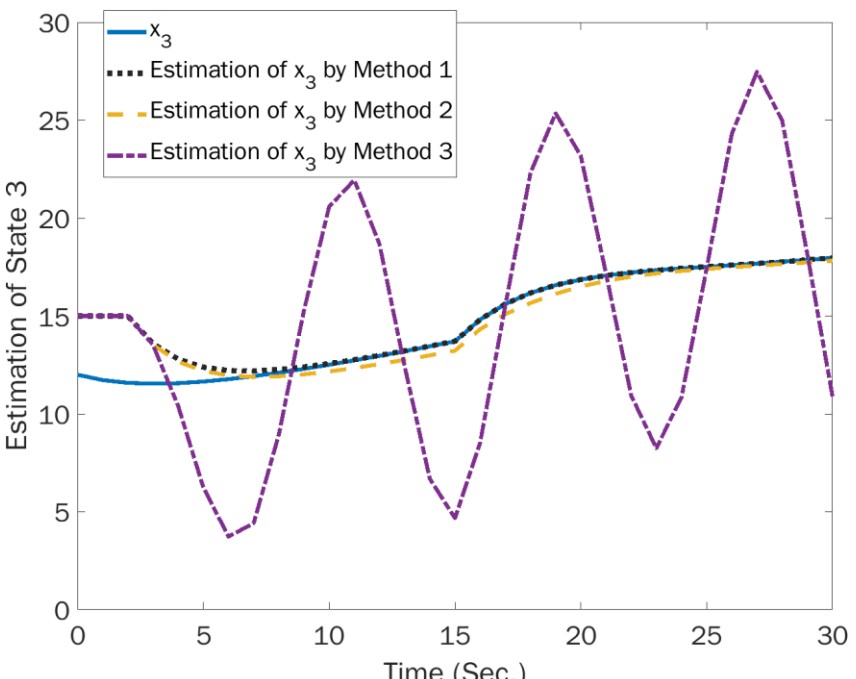

**Figure 5.** The state 3 estimation results of the three introduced observers.

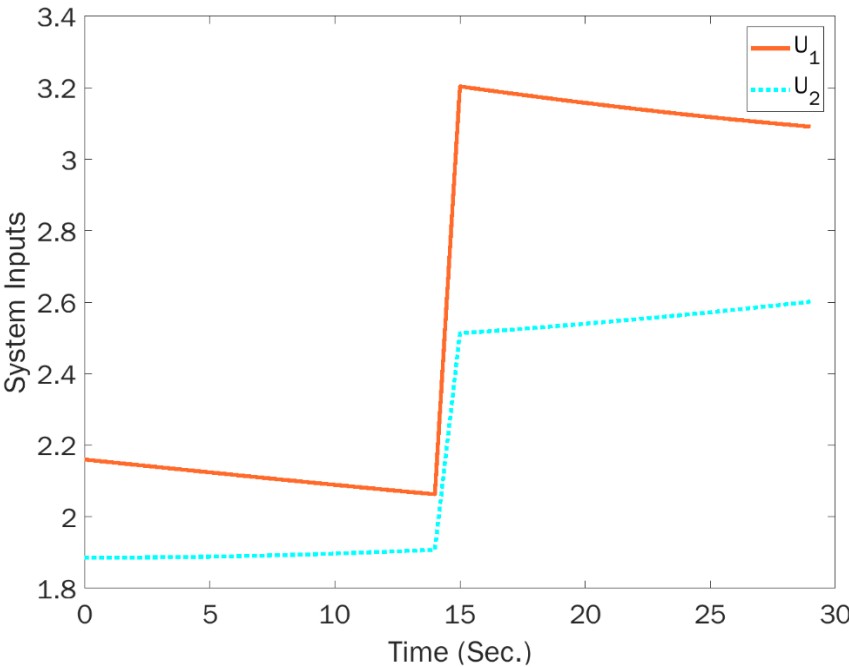

**Figure 6.** Shows inputs signals of the CSTH system.

**Remark 3**. *Based on the results presented above, we can conclude that*

- *By changing the observer poles, the convergence speed can be tuned.*
- *Each chain is an observer in the form of (6).*
- *Increasing the communication delays may increase the convergence time and transient estimation error magnitude, but according to theorem 3, it will not make the observer unstable.*

## 6. Conclusions

Delays are an inherent part of any system. Ignoring it could cause instability or degradation in the performances of the control or estimation blocks. Measurement and communication lines produce some delays, which may be unequal due to different line distances or sensor types and technologies. Supposing equal measurement/communication delays causes a similar problem to ignoring them. In this paper, we proposed a new approach to constructing an observer, consisting of a bank of chain-observers to deal with equal and unequal measurement delays. Analytical results for the construction of the observers are provided, and convergence of the chain–observers is shown. Simulations demonstrate the efficacy of the proposed design scheme through a CSTH system in the presence of measurement delays. Having a chain-based observer design facilitates the extension of the proposed approach to a data-driven method. Therefore, the advantage of the proposed approach is that it can be used for designing data-driven (subspace-based) observers [17] in the presence of equal and unequal communication delays. Designing such a data-driven observer is in our future research scope.

**Author Contributions:** Conceptualization, Y.A. and L.Z.; Methodology, Y.A. and L.Z.; Software, Y.A.; Validation, Y.A. and L.Z.; Formal analysis, Y.A.; Investigation, Y.A.; Resources, Y.A. and L.Z.; Data curation, Y.A.; Writing—original draft preparation, Y.A.; Writing—review and editing, Y.A. and L.Z.; Visualization, Y.A.; Supervision, L.Z.; Project administration, L.Z.; Funding acquisition, L.Z. All authors have read and agreed to the published version of the manuscript.

**Funding:** This research is supported by Alberta Innovates with award number 202102493.

**Data Availability Statement:** Data sharing is not applicable (no new data were created or analyzed in this study, therefore, data sharing was not applicable to this article).

**Acknowledgments:** The authors gratefully acknowledge the financial support from Alberta Innovates.

**Conflicts of Interest:** The authors declare no conflict of interest.

## Appendix A. Proof of Theorem 2

Before proving Theorem 2, the following lemma is presented.

**Lemma A1.** *For any two matrices F and G of dimension $n \times m$ and for any square matrix $P = P^T > 0$ of dimension n, the following matrix inequality holds,*

$$(F+G)^T P(F+G) \leq (1+\varepsilon)F^T PF + \left(1+\varepsilon^{-1}\right)G^T PG \tag{A1}$$

*where $\varepsilon$ is a positive constant.*

Now, to prove Theorem 2, consider the following Lyapunov function:

$$V(x(k)) = x(k)^T Px(k) + \sum_{j=1}^{N}\sum_{l=1}^{d_j} x^T(k-l)S_j x(k-l)$$
$$P = P^T > 0$$
$$S_j = S_j^T \geq 0 \tag{A2}$$

Taking forward the difference from the above equation gives:

$$\Delta V(x(k))\, V(x(k+1)) - V(x(k))$$

$$= \left[A_0 x(k) + \sum_{j=1}^{N} A_j x(k-d_j)\right]^T P \left[A_0 x(k) + \sum_{j=1}^{N} A_j x(k-d_j)\right]$$

$$-x(k)^T P x(k) + \sum_{j=1}^{N}\sum_{l=1}^{d_j} x^T(k+1-l) S_j x(k+1-l)$$

$$-\sum_{j=1}^{N}\sum_{l=1}^{d_j} x^T(k-l) S_j x(k-l)$$

$$P = P^T > 0$$

$$S_j = S_j^T \geq 0 \tag{A3}$$

Applying Lemma A1 on inequality (A3) yields

$$\Delta V(x_k) \leq (1+\varepsilon) x^T(k) A_0^T P A_0 x(k) +$$

$$+(1+\varepsilon^{-1}) \sum_{j=1}^{N} x^T(k-d_j) A_j^T P \sum_{j=1}^{N} A_j x(k-d_j)$$

$$-x^T(k) P x(k) + x^T(k) \sum_{j=1}^{N} S_j x(k) \tag{A4}$$

$$-\sum_{j=1}^{N} x^T(k-d_j) S_j x(k-d_j)$$

Now, based on Cauchy-Schwarz inequality [22], namely for any real vector $v_i$, the inequality $\left(\sum_{i=1}^{m} v_i\right)^T \left(\sum_{i=1}^{m} v_i\right) \leq m \sum_{i=1}^{m} v_i^T v_i$ holds, and we have

$$\Delta V(x_k) \leq x^T(k)\left[(1+\varepsilon) A_0^T P A_0 + \sum_{j=1}^{N} S_j - P\right] x(k)$$

$$+(1+\varepsilon^{-1}) N \sum_{j=1}^{N} x^T(k-d_j) A_j^T P A_j x(k-d_j) \tag{A5}$$

$$-\sum_{j=1}^{N} x^T(k-d_j) S_j x(k-d_j)$$

Taking two factors $x^T(k-d_j)$ and $x(k-d_j)$ out of the left- and right-hand side of two last terms in Equation (A5) yields

$$\Delta V(x_k) \leq x^T(k)\left[(1+\varepsilon) A_0^T P A_0 + \sum_{j=1}^{N} S_j - P\right] x(k)$$

$$+\sum_{j=1}^{N} x^T(k-d_j)\left[N(1+\varepsilon^{-1}) A_j^T P A_j - S_j\right] x(k-d_j) \tag{A6}$$

Now, select $S_j = N(1+\varepsilon^{-1}) A_j^T P A_j$. We have

$$\Delta V(x_k) \leq x^T(k)\Big[(1+\varepsilon) A_0^T P A_0$$

$$+N(1+\varepsilon^{-1}) \sum_{j=1}^{N} A_j^T P A_j - P\Big] x(k) \tag{A7}$$

$$= \varphi(x,\varepsilon)$$

Since matrices $A_0^T P A_0$ and $A_j^T P A_j$ are symmetric and positive semi-definite, based on [23], we have

$$\begin{aligned} \varphi(x, \varepsilon) &\leq x^T(k)\Big[(1+\varepsilon)\lambda_{\max}\big(A_0^T P A_0\big) \\ &+ N\big(1+\varepsilon^{-1}\big)\sum_{j=1}^{N}\lambda_{\max}\big(A_j^T P A_j\big) - P\Big]x(k) \\ &= g(\varepsilon)\lambda_{\max}(P)\|x(k)\|_2^2 \end{aligned} \tag{A8}$$

where

$$g(\varepsilon) = (1+\varepsilon)\sigma_{\max}^2(A_0) + N\Big(1+\varepsilon^{-1}\Big)\sum_{j=1}^{N}\sigma_{\max}^2(A_j) \tag{A9}$$

The minimum of the function $g$ can be found by taking its derivative with respect to the parameter $\varepsilon$, which yields

$$\frac{dg(\varepsilon)}{d\varepsilon} = 0 \;\Rightarrow\; \sigma_{\max}^2(A_0) - \frac{N}{\varepsilon^2}\sum_{j=1}^{N}\sigma_{\max}^2(A_j) = 0 \tag{A10}$$

Therefore, the optimum value for $\varepsilon$ is

$$\varepsilon_m = \left(N\sum_{j=1}^{N}\sigma_{\max}^2(A_j)\right)^{\frac{1}{2}}\sigma_{\max}^{-1}(A_0) \tag{A11}$$

We can conclude

$$\begin{aligned} \Delta V(x_k) &\leq \varphi(\varepsilon_m, x_k) \\ &= x^T(k)\Big[(1+\varepsilon_m)\lambda_{\max}\big(A_0^T P A_0\big) \\ &+ N\big(1+\varepsilon_m^{-1}\big)\sum_{j=1}^{N}\lambda_{\max}\big(A_j^T P A_j\big) - P\Big]x(k) \end{aligned} \tag{A12}$$

Now, if the condition $(1+\varepsilon_m)A_0^T P A_0 + N\left(\frac{1+\varepsilon_m}{\varepsilon_m}\right)\sum_{j=1}^{N}A_j^T P A_j - P < 0$ is satisfied, then considering Equation (A8), we have

$$\Delta V(x_k) \leq -\beta\|x(k)\|_2^2 \tag{A13}$$

The proof is completed.

**Appendix B. Proof of Theorem 3**

From Equations (1) and (27), we have

$$y(k) = Cx(k) + Du(k) \tag{A14}$$

and

$$\begin{aligned} \hat{y}(k) &= W_i z(k-\mathbf{d}) + V_i \bar{y}(k-\mathbf{d}) \\ &+ \sum_{i=0}^{d_i} Q_i u(k-i) \end{aligned} \tag{A15}$$

Substituting $x(k)$ from Equation (A14) in (A15), gives

$$y(k) = CA^d x(k-d) + \sum_{i=0}^{d-1} CA^i B u(k-1-i) + Du(k) \tag{A16}$$

Now, subtracting (A16) from (A15), gives

$$y(k) - \hat{y}(k) = CA^d x(k-d) + \sum_{i=0}^{d-1} CA^i Bu(k-1-i) + Du(k) - W_i z(k-\mathrm{d}) + V_i \overline{y}(k-\mathrm{d}) - \sum_{i=0}^{d_i} Q_i u(k-i) \quad \text{(A17)}$$

or

$$
\begin{aligned}
y(k - \tau_{i-1}) &- \hat{y}(k - \tau_{i-1}) \\
&= CA^d x(k - d - \tau_{i-1}) + \sum_{i=0}^{d-1} CA^i Bu(k-1-i-\tau_{i-1}) \\
&+ Du(k - \tau_{i-1}) - W_i z(k - \mathrm{d} - \tau_{i-1}) - V_i \overline{y}(k - \mathrm{d} - \tau_{i-1}) \\
&\quad - \sum_{i=0}^{d_i} Q_i u(k - i - \tau_{i-1})
\end{aligned}
\quad \text{(A18)}
$$

Now, recalling that $\tau_i = d + \tau_{i-1}$, the above equation can be rewritten as

$$
\begin{aligned}
y(k - \tau_{i-1}) &- \hat{y}(k - \tau_{i-1}) \\
&= CA^d x(k - \tau_i) + \sum_{i=0}^{d-1} CA^i Bu(k-1-i-\tau_{i-1}) \\
&+ Du(k - \tau_{i-1}) - W_i z(k - \tau_i) - V_i \overline{y}(k - \tau_i) \\
&\quad - \sum_{i=0}^{d_i} Q_i u(k - i - \tau_{i-1})
\end{aligned}
\quad \text{(A19)}
$$

Now, according to Equation (10), for chain $i$, we can write

$$
\begin{aligned}
\hat{y}(k - \tau_{i-1}) &= y(k - \tau_{i-1}) - W_i e(k - d_i) \\
&for \ i = 1, \ldots, m
\end{aligned}
\quad \text{(A20)}
$$

Then, we have

$$\overline{y}(k - \tau_{i-1}) = y(k - \tau_{i-1}) - W_i I_i e(k - d_i). \quad \text{(A21)}$$

From (A19), for each chain $i$, we get

$$
\begin{aligned}
y(k - \tau_{i-1}) - \hat{y}(k - \tau_{i-1}) &= CA^{d_i} x(k - \tau_i) \\
+ \sum_{j=0}^{d_i-1} CA^j Bu(k - j - \tau_{i-1}) &+ Du(k - \tau_{i-1}) \\
-W_{i-1} z(k - \tau_i) &- V_{i-1} \overline{y}(k - \tau_{i-1}) \\
- \sum_{j=0}^{d_i-1} Q_j u(k - j - \tau_{i-1}) &
\end{aligned}
\quad . \quad \text{(A22)}
$$

Substituting Equation (A21) in the above equation yields

$$
\begin{aligned}
y(k - \tau_{i-2}) - \hat{y}(k - \tau_{i-2}) &= CA^{d_{i-1}} x(k - \tau_{i-1}) \\
+ \sum_{j=d_{i-1}}^{d_{i-2}-1} CA^j Bu(k - j - \tau_{i-1}) &+ Du(k - \tau_{i-2}) \\
-W_{i-1} z(k - \tau_{i-1}) &- V_{i-1} y(k - \tau_{i-1}) \\
-V_{i-1} W_i I_i e(k - \tau_i) &- \sum_{j=d_{i-1}}^{d_{i-2}-1} Q_j u(k - j - \tau_{i-2})
\end{aligned}
\quad \text{(A23)}
$$

Now, selecting $CA^{d_{i-1}} = W_{i-1} T_{i-1} + VC$, $Q_{d_{i-1}} = D - V_{i-1} D$, $Q_j = CA^j B$, $j = d_i + 1, , \ldots, d_{i-1}$ gives

$$
\begin{aligned}
y(k - \tau_{i-2}) - \hat{y}(k - \tau_{i-2}) &= V_{i-1} W_i e(k - \tau_i) \\
&+ W_{i-1} e(k - \tau_{i-1})
\end{aligned}
\quad \text{(A24)}
$$

Then,

$$\bar{y}(k - \tau_{i-1}) = y(k - \tau_{i-1}) - V_{i-1}W_i I_i e(k - \tau_i) \\ - W_{i-1} I_{i-1} e(k - \tau_{i-1}) \tag{A25}$$

Comparing Equations (A20) and (A25), and following similar steps from (A20) to (A23) on (A25), we will reach:

$$y(k - \tau_{i-3}) - \hat{y}(k - \tau_{i-3}) = V_{i-2}V_{i-1}W_i e(k - \tau_i) + V_{i-1}W_{i-1}e(k - \tau_{i-1}) + + W_{i-2}e(k - \tau_{i-2}) \tag{A26}$$

Repeating this sequence until chain $m$, gives

$$y(k - \tau_i) - \hat{y}(k - \tau_i) = \sum_{j=i+1}^{m} \prod_{p=m}^{j} V_p W_j e(k - \tau_j) \tag{A27}$$

and

$$\bar{y}(k - \tau_{i-1}) = y(k - \tau_{i-1}) - \sum_{j=i+1}^{m} \prod_{p=m}^{j} V_p W_j I_j e(k - \tau_j) \tag{A28}$$

According to Equation (A27), the convergence of states brings

$$\lim_{k \to \infty} y(k - \tau_i) - \hat{y}(k - \tau_i) = 0 \tag{A29}$$

Now, the estimation error can be written as

$$e(k + 1) = T_i x(k + 1) - z(k + 1) \tag{A30}$$

Substituting Equations (A22) and (26) in Equation (A30) gives

$$e(k + 1) = T_m A^{d_m + 1} x(k - \tau_m) \\ + \sum_{j=0}^{d_i} T_i A^j B u(k - j - \tau_m) - G_m z(k - \tau_m) \\ - \sum_{j=0}^{d_i} H_j u(k - j - \tau_m) - L_i \bar{y}(k - \tau_m) \tag{A31}$$

Now, substituting $T_i A^{d_i} - G_i T_i = L_i C$, $H_{d_i - 1} = T_i A^{d_i - 1} B - L_i D$, $H_j = T_i A^j B$, $j = 0, \ldots, d_i - 2$ and considering Equation (26) to (29), the above equation is reduced to

$$e(k + 1) = G_m e(k - \tau_m) - L_i \sum_{j=i+1}^{m} \prod_{p=m}^{j} V_p W_j I_j e(k - \tau_j) \tag{A32}$$

From Equation (18) in Theorem 1 and Remark 2, we know that

$$e(k) = G e(k - d)$$

Considering the stability of G, we have $\lim_{k \to \infty} e(k) = 0$. Then, considering all matrices in Equation (A32) are constant, we can conclude

$$\lim_{k \to \infty} y(k - \tau_i) - \hat{y}(k - \tau_i) = \lim_{k \to \infty} e(k - \tau_i) \to 0$$

Therefore, the estimation error will converge to zero, provided that the error dynamics (A32) is stable (i.e., condition (29) is satisfied).

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
