# Peer review of "Model-Based Observer Design Considering Unequal Measurement Delays"

_actuators, doi:10.3390/act10110281_

Round 1

Reviewer 1 Report

The system (4) is not found as such in reference (13). The system (4) has 4 equations, and the unknowns are 5. How do the authors comment on the issue?

Reviewer 3 Report

See the file

Reviewer 4 Report

Modern control systems of linear and nonlinear phenomena based on mathematical modeling of processes are an important part of control theory and applications. In fact, several studies have appeared in the literature concerning the extension or the adaptation of classic results on fundamental control problems to a vast class of systems, in particular, those involving state observers. Thus, the paper under review is relevant to the field and of particular scientific importance.

The paper discusses the possibility of implementing a design of observers with consideration of measurement delays for both equal and unequal measurement delays in outputs and a chain of observers is proposed. Overall state estimation and convergence are proved and some simulations presented.

In general, the paper is well-written and easy to follow. However, I have some questions and suggestions for the authors. Please, find my remarks below.

  1. First, clarify the scientific contribution of your study. The authors state that the measurement delay is often ignored by published papers. Nonetheless, in general, all authors claim about the efficiency of their approaches when discuss their results. In this paper, is the proposed approach, which uses "measurement delays " really novel ? (by the way, what kind of novelty? Maybe it is of reason to clarify it more forcefully (including in the Abstract?). I disagree with the authors when they state that, "in the existing state estimation methods, the measurement delay is often ignored". There are well written papers considering measurement delay. Please provide further clarification about the contribution of the current paper.
  2. How the efficiency of handling parametric uncertainties was evaluated in your study?
  3. Sections 3, 4 and 5 could be merged into a Section (plus sub-sections), i.e., I suggest that the current Sections 4 and 5 change to sub-sections 3.1 and 3.2, respectively.
  4. The authors state that simulations were performed using CSTH system. Nonetheless, more detailed information should be provided, e.g., how the proposed algorithm was implemented to generate Figs. 3-6 (a modified version of CSTH system) ? Please clarify.
  5. What is the computational complexity of the proposed controller? Could it be implemented as embedded software? Or what about real-time control?
  6. A performance comparison (including statistical treatment) between the proposed method and existing apporaches published in the literature is imperative in order to certify de efficiency of the method proposed by authors in this study.
  7. The Conclusion section could be expanded with a more detailed description of the achieved results.
  8. Additional proofreading is needed to fix some typos.

Round 2

Reviewer 2 Report

It can be accepted.

Author Response

Thanks.

Reviewer 3 Report

My comments are attached

Author Response

Thank you for your comments. Attached pls see our response.

Reviewer 4 Report

I appreciate the effort made by authors in addressing my observations. The paper has improved and the authors managed to address my concerns.

Author Response

Thanks.